# Multi-Target-Directed Cinnamic Acid Hybrids Targeting Alzheimer’s Disease

**DOI:** 10.3390/ijms25010582

**Published:** 2024-01-01

**Authors:** Aliki Drakontaeidi, Eleni Pontiki

**Affiliations:** Department of Pharmaceutical Chemistry, School of Pharmacy, Faculty of Health Sciences, Aristotle University of Thessaloniki, 54124 Thessaloniki, Greece; alikdrak@pharm.auth.gr

**Keywords:** Alzheimer’s disease, cinnamic acids, hybrids, acetylcholinesterase inhibition, multi-target, antioxidant, neuroprotective

## Abstract

Progressive cognitive decline in Alzheimer’s disease (AD) is a growing challenge. Present therapies are based on acetylcholinesterase inhibition providing only temporary relief. Promising alternatives include butyrylcholinesterase (BuChE) inhibitors, multi-target ligands (MTDLs) that address the multi-factorial nature of AD, and compounds that target oxidative stress and inflammation. Cinnamate derivatives, known for their neuroprotective properties, show potential when combined with established AD agents, demonstrating improved efficacy. They are being positioned as potential AD therapeutic leads due to their ability to inhibit Aβ accumulation and provide neuroprotection. This article highlights the remarkable potential of cinnamic acid as a basic structure that is easily adaptable and combinable to different active groups in the struggle against Alzheimer’s disease. Compounds with a methoxy substitution at the para-position of cinnamic acid display increased efficacy, whereas electron-withdrawing groups are generally more effective. The effect of the molecular volume is worthy of further investigation.

## 1. Introduction

Alzheimer’s disease (AD), a neurodegenerative disorder characterized by progressive cognitive decline, is expected to become increasingly prevalent in the coming years, according to researchers [1,2,3,4]. It is also worth noting that Alzheimer’s disease is recognized as the most common late-life mental disorder [5]. The fundamental pathology of the disease involves the loss of nerve cell function, primarily impacting areas of the brain linked to memory and cognition [6]. Despite extensive research, a definitive cause or etiology for the disease remains elusive. However, several common features are associated with Alzheimer’s disease, including reduced levels of acetylcholine (ACh), aggregation of amyloid-β (Aβ) deposits and increased oxidative stress. Extensive hyperphosphorylation of tau tangles and disorders in the homeostasis of certain biometallic ions are also hallmarks of the disease [7,8,9,10,11,12,13,14,15,16]. The condition is also associated with brain-derived neurotrophic factor, cell cycle and mitochondrial dysfunction hypotheses [17,18,19,20].

The pathogenic effects of Aβ accumulation are closely linked to lipid membrane permeability, oxidative stress and mitochondrial and endoplasmic reticulum dysfunction. In addition, excessive phosphorylation of the tau protein can disrupt both microtubule function and intercellular transport processes, while extracellular Aβ accumulation can lead to neuroinflammation by triggering the activation of microglia and astrocytes [21,22,23,24,25]. Treatments targeting this mode of pathophysiology typically involve secretase inhibitors or Aβ antibodies. Secretases are enzymes responsible for processing the amyloid precursor protein (APP) into both non-amyloidogenic and amyloidogenic fragments, making them critical for Aβ generation. However, these therapeutic approaches have not been as effective, often due to side effects or difficulties in reaching their intended targets [26,27,28,29].

Up to now, the treatment of the disease has mainly revolved around drugs that aim to increase acetylcholine levels by inhibiting the enzyme acetylcholinesterase (AChE). This is due to the association of AD with the loss of cholinergic function. There are several drugs that have been used for this purpose, including rivastigmine, donepezil, galantamine and tacrine (Figure 1) [30,31,32,33,34,35,36,37,38,39,40]. Except for tacrine, all of these agents are still in active use within clinical settings. Tacrine was withdrawn from the market due to hepatotoxicity [41,42,43]. In some cases, N-methyl-D-aspartate (NMDA) antagonists, which regulate glutamate activity by inhibiting NMDA receptors, have been used to relieve some of the disease’s symptoms [44]. While these drugs provide relief from the symptoms of the disease and improve the patient’s general condition, including cognitive function and memory, it is important to note that their effectiveness is temporary [30,31,32,33,34,35,36,37,38,39,40,45,46].

It is important to distinguish between the two types of cholinesterase (ChE) associated with Alzheimer’s disease (AD): acetylcholinesterase (AChE) and butyrylcholinesterase (BuChE). Under normal conditions, the metabolism of acetylcholine is mainly controlled by AchE. However, in advanced Alzheimer’s disease, AchE activity in the brain decreases dramatically, sometimes by as much as 90%, while BuChE activity remains relatively stable [47,48,49,50]. In addition, a number of research studies have shown that BuChE inhibitors have a longer duration of action, increased stability and several other advantages in the treatment of Alzheimer’s disease [51]. Consequently, inhibition of BuChE has emerged as a promising therapeutic strategy and has been extensively investigated. Simultaneous inhibition of both enzymes also appears to be a potential strategy in the quest for Alzheimer’s disease (AD) therapies [52,53].

It is worth noting that the two enzymes share 65% amino acid homology and have two binding sites for inhibitors: the catalytic active site (CAS), located at the base of the enzyme, and the peripheral anionic site (PAS), located near the gorge entrance (Figure 2). Designing molecules that target both sites has been a widely investigated approach in recent years [54,55].

Another enzyme that has been identified playing an important role in the pathophysiology of Alzheimer’s disease (AD) is monoamine oxidase B (MAO-B). Recent studies suggest increased MAO-B expression in Alzheimer’s patients in brain regions such as the hippocampus and cerebral cortex [57]. MAO-B is involved in the generation of hydroxyl radicals in the brain, which promote the accumulation of Aβ, a hallmark of AD pathology. Therefore, inhibition of MAO-B is considered a promising avenue for the treatment of AD. Selegiline (Figure 3), a known MAO-B inhibitor, has been demonstrated to improve cognition for people with AD [58,59].

Certain cellular processes occur before the onset of other features of Alzheimer’s disease (AD). Microglia and resident astrocytes trigger the production of pro-inflammatory cytokines such as NO, TNFα and ROS, which are a critical aspect of the inflammatory response associated with AD [60,61,62]. This phenomenon takes place before the appearance of amyloid beta plaques and neurofibrillary tangles. Evidence from many studies suggests that substances with antioxidant and anti-inflammatory properties may alleviate Alzheimer’s symptoms and inhibit disease progression. As a result, carefully designed drugs with antioxidant and anti-inflammatory properties may prove essential in both the prevention and treatment of Alzheimer’s disease [63,64].

To improve the treatment of Alzheimer’s disease (AD), recent research has focused on the design and synthesis of multi-target-directed ligands (MTDLs), which have shown considerable promise [65,66,67,68,69,70,71,72]. This is due to the multifactorial nature of Alzheimer’s disease. Single-target therapeutics for the treatment of Alzheimer’s disease have been rejected from clinical trials in recent years due to the complex and numerous molecular processes involved in the disease [72,73,74,75]. It is noteworthy that many therapeutic agents have been synthesized to target multiple biological entities simultaneously, including AChE inhibitors, inhibitors of Aβ (amyloid-beta) and tau protein accumulation, monoamine oxidase inhibitors and compounds with antioxidant activity [72,76,77,78,79,80,81,82,83,84,85]. The mentioned approach also improves patient compliance by minimizing the necessity of taking a combination of drugs [71].

Cinnamic acid and its derivatives, found in many plants, have been extensively researched for their antioxidant, anti-inflammatory and neuroprotective properties [86,87,88]. Given the established link between Alzheimer’s disease (AD) and oxidative stress, the administration of antioxidants would appear to be beneficial. However, studies have shown that the efficacy of antioxidants alone is significantly lower than their efficacy when in hybrids with recognized AD agents [89,90,91,92]. In addition, molecular hybridization has emerged as a prominent strategy in recent years, involving the combination of different bioactive groups in one single molecule to optimize therapeutic action and efficacy [93]. Moreover, research suggests that certain cinnamic acid derivatives have the ability to inhibit Aβ accumulation in vitro and have neuroprotective properties [94]. As a result, cinnamic acid is considered a potential lead compound for the development of agents for the management of neurodegenerative diseases such as Alzheimer’s disease [95].

The main objective of this review is to summarize recent advances in the use of cinnamic acid derivatives synthesized by hybridization with established bioactive moieties for the treatment of Alzheimer’s disease (AD). Furthermore, this review aims to analyze the results of these studies and to highlight the multifaceted activities of these compounds as multi-target agents.

## 2. Results

### 2.1. Tacrine Derivatives

Chen, Y. et al. [96] conducted a study involving the synthesis of tacrine (Figure 1) analogs fused to cinnamic acid derivatives. The compounds were designed on the basis that cinnamic acid derivatives have previously shown activity against Alzheimer’s disease. It was hypothesized that these compounds could access the narrow pocket within the active site of the AChE enzyme and develop interactions with the peripheral anionic site (PAS). The primary objective of this research was to evaluate the inhibitory potential of these compounds against two enzymes, namely AChE (from *Electrophorus electricus*) and BuChE (from equine serum). The study followed established protocols for in vitro assessment of enzymatic activity using Ellman’s method [66,97]. In addition, enzyme kinetics [98] and inhibitory effects on self-induced Ab1-42 aggregation [99] were investigated.

To achieve this, a series of tacrine-based molecules were synthesized, each attached to cinnamic acid via a diamine linker chain, varying in the number of intermediate carbons. Compounds with a higher number of intermediate carbons, especially those with six and eight carbon atoms, showed greater efficacy against AChE, which is confirmed by biological experiments. Among these two compounds, the one with a 6-carbon linker was the most favorable due to its comparable activity and lower molecular weight.

Having established the general structure (Figure 4) of the compounds, the researchers investigated different substituents on cinnamic acid in different positions. Methyl substitution increased anti-AChE activity when positioned at the para position only. Nitro substitution generally increased activity, with the para-nitro-substituted compound showing the highest activity. The -CF_3_ substitution increased activity, but only when placed at the para position. In summary, these substitutions were most effective in inhibiting AChE when located at the para position, followed by the meta position, and least effective at the ortho position. The pattern was reversed for the effect on BuChE. Halogen substitution had a stronger effect with 4-chloro substitution, followed by 4-bromo and then the less potent 4-fluoro substitution [65,66,70]. This pattern was consistent for both enzymes. The addition of a 4-OCH_3_ group increased the activity of both enzymes. The triple substituted 3,4,5-tri-OCH_3_ molecule displayed significant efficacy, surpassing 2,3,4-tri-OCH_3_. Compounds with a hydroxy group at positions 3 and 4 exhibited selectivity over AChE. Derivatives with a benzyloxy group in the para position showed reduced activity against AChE, while retaining activity against BuChE. This may be due to the broader binding site of BuChE, allowing these bulkier molecules to bind effectively in contrast to the narrower region of AChE. The compounds featuring 4-nitro, 4-chloro and 4-methoxy substitutions were also tested for their ability to inhibit the human enzymes (AChE and BuChE), to prevent self-induced Ab1-42 aggregation and to improve cognitive performance in the Morris water maze test. These results confirmed the potential of these compounds as promising multi-target therapeutics for the treatment of Alzheimer’s disease.

Quintanova, C. and colleagues [100] synthesized and studied tacrine hybrids combined with cinnamic acid and cinnamylidene acetic acid. Their primary aim was to enhance the antioxidant properties by extending the conjugation of the double bond found in cinnamic acid. Each compound was screened for activity against AChE using the Ellman protocol [97]. Three compounds were synthesized for each tested substituent on the cinnamate moiety, with carbon numbers between tacrine and cinnamic acid ranging from two to four. The most promising compounds showed a 3,4-dimethoxy substitution and had a carbon number of two or three in the linker between the tacrine and the cinnamate residue (compounds I and II, Figure 5). Subsequently, compounds containing a methylenedioxy substituent with a linker carbon number of two or three showed considerable potential (compounds III and IV, Figure 5). Previous molecular docking studies by these researchers suggested that non-polar substituents facilitate hydrophobic interactions, thereby increasing the stability of their binding to the enzyme. Conversely, compounds with polar substituents, such as hydroxyl, showed reduced activity.

In addition, these compounds were subjected to an antioxidant capacity test involving interaction with the stable free radical 2,2-diphenyl-1-picrylhydrazyl (DPPH) [101]. It was observed that compounds containing hydroxyl groups exhibited higher antioxidant capabilities compared to the others, which showed moderate effects. In addition, these compounds were screened for inhibition of Aβ accumulation in vitro using thioflavin T (ThT). They were also tested in neuronal cells subjected to Aβ-induced stress and hydrogen peroxide (H_2_O_2_) to evaluate their potential neuroprotective role. These results indicate that derivatives similar to these compounds may be used as multipotent agents to halt Alzheimer’s disease (AD).

### 2.2. Isoquinoline and Quinoline Derivatives

Building on their previous research, Wang K. and colleagues [102], who had previously demonstrated the inhibitory potential of the 1,2,3,4-tetrahydroisoquinoline group against the BuChE enzyme [103,104,105], developed hybrid compounds by combining this group with various cinnamic acids. These compounds were screened as inhibitors of AChE and BuChE enzymes using the Ellman method [97] with donepezil as the reference compound.

The results showed that the synthesized hybrid compounds were not effective against AChE, but some demonstrated significant inhibitory activity against BuChE. In particular, the unsubstituted cinnamic acid-derived molecule (compound V—Figure 6) exhibited potent activity against BuChE, and this activity was significantly enhanced when the molecule was substituted with a hydroxy group (compound VI—Figure 6). Similarly, when the benzene ring of the acid was replaced with thiophene, the activity increased significantly (compound VII—Figure 6). However, the introduction of fluorine or the replacement of the benzene ring by naphthalene, pyridine or other acids such as *(E)*-2-methyl-3-phenylacrylic acid, 3-phenylpropiolic acid or methacrylic acid resulted in reduced activity against the BuChE enzyme. Reducing the double bond also decreased the activity of the compounds. Compound VII significantly inhibited human BuChE making it a potent and reversible inhibitor of the enzyme, which was proven through kinetic studies. In addition, both compounds VI and VII exhibited a remarkable inhibitory activity against monoamine oxidase B (MAO-B) and was shown to have antioxidant properties through the oxygen radical absorbance capacity fluorescein (ORAC-FL) assay [106], applying Trolox as a reference compound. Compound VII also displayed anti-inflammatory properties on LPS-induced PC12 cell injury (via the MTT assay [107]) and neuroprotective properties against SH-SY5Y cell injury induced by Aβ, via the MTT assay [108]. Furthermore, compound VII improved dyskinesia recovery rates and response efficiency in an AlCl3-induced zebrafish model. It further ameliorated scopolamine-induced memory impairment.

Quinoline, a pharmacophore group, has been extensively used in the synthesis of bioactive compounds against neurodegenerative diseases. In addition, certain quinoline derivatives have been proposed as potential anti-AD agents [109,110]. Ge, Y.X. et al. [111] undertook the synthesis and evaluation of hybrid quinoline derivative molecules combined with cinnamic acids. They evaluated the efficacy of the compounds in inhibiting Aβ accumulation using the thioflavin T (ThT) fluorescence method [112]. In this study, resveratrol was used as a positive control and galantamine as a negative control. Compounds VIII, IX and X (Figure 7) showed similar activity to resveratrol and this activity was found to be dose dependent. Interestingly, the addition of substituents that are electron withdrawing groups, such as halogens, to the cinnamic acid moiety resulted in decreased activity or increased accumulation of Aβ.

In addition, promising compounds were evaluated for cytotoxicity against SHSY5Y cells using the MTT method [113]. All compounds exhibited IC_50_ values greater than 50 μM. Toxicity against the human hepatocyte cell line (LO2) was also evaluated, with IC_50_ values between 40 and 60 μM. The compounds were also evaluated for their activity against Aβ_42_-induced neurotoxicity in SH-SY5Y cells, using EGCG as a positive reference compound. The results showed a dose-dependent increase in the viability of SH-SY5Y cells. Based on these observations, it was concluded that these compounds have potential as neuroprotective agents and could serve as therapeutic candidates against Alzheimer’s disease.

Building on their previous research [99,114] on the synthesis of cinnamate derivatives of tacrine [15,16], Mo, J. et al. [115] started to develop new molecules by removing the cyclohexane ring from tacrine. These efforts led to the synthesis of quinoline derivatives hybridized with ferulic acid (Figure 8), which allowed further investigation of the structure–activity relationship. Interestingly, these new compounds exhibited a reduced molecular weight.

In this case, the substituents are located on the terminal benzene ring of the ferulic acid, rather than on the benzene moiety of the cinnamic acid. Following the Ellman [97] protocol, the synthesized compounds were screened for their inhibitory activity against AChE and BuChE. Comparative analysis showed that the compound with a methyl group at the para position had activity similar to that of the unsubstituted compound. The compound with fluorine in the para position showed selectivity against BuChE, whereas the chloro substitution in the same position showed selectivity against AChE. Increased activity against both enzymes was observed for the compound with boron in the para position. Similarly, other electron withdrawing substituents such as 4-CF_3_ and 4-CN consistently demonstrated improved activity.

They subsequently synthesized quinoline derivatives with cinnamic acid (Figure 8), which gave comparable results in terms of substituents. The fluorine-substituted derivative also showed reduced activity, while the molecule with the highest activity against AChE had a nitro substitution. In general, electron-withdrawing substituents increased the activity of these molecules.

The two most promising compounds (Figure 9) underwent kinetic studies using Lineweaver–Burk plots [97], demonstrating a competitive type of inhibition. In addition, MTT assays performed on the HepG2 cell line [116] revealed hepatotoxicity, with compound XI showing higher toxicity compared to tacrine, while compound XII showed lower toxicity. A neuroprotective effect against H_2_O_2_-induced cell death was also detected in PC12 neurons, with compound XI showing a better neuroprotective effect. However, neither compound showed antioxidant properties in a 1,1-diphenyl-2-picrylhydrazyl (DPPH) [117] radical scavenging assay. They also failed to inhibit Aβ amyloid accumulation in a thioflavin T-based fluorimetric assay. Notably, compound XII showed a low antioxidant capacity of 27%. Consequently, the study suggests further investigation into ring replacement with alternative heterocycles.

### 2.3. Benzylpiperidine Derivatives 

The *N*-benzylpiperidine group, present in established AChE inhibitors such as donepezil (Figure 1) [118,119], served as the basis for hybrid molecules synthesized by Martín Estrada et al. [120]. These hybrids incorporated *N*-benzylpiperidine and cinnamic acid. The compounds were evaluated for their inhibitory activity on human cholinesterases (hAChE and hBuChE) using the Ellman method [97], with donepezil as the reference compound. The most active compound, illustrated in Figure 10, showed greater selectivity towards BuChE than other synthesized derivatives. In particular, the methoxy group on the benzene ring of cinnamate was found to contribute significantly to the activity of the compound.

However, this methoxy group contribution did not extend to inhibition of human MAO-A and MAO-B enzymes. In this context, the methoxy derivatives were almost inactive, whereas the p-hydroxy derivatives showed better inhibition. Interestingly, the di-hydroxy derivatives were the most effective. The compound with the most potential also showed good antioxidant properties in the ORAC assay. It also demonstrated neuroprotective properties, as it was able to protect the human neuroblastoma cell line SH-SY5Y against damage by free radicals in the mitochondria. 

Based on their previous research, Wang, K. [102] and colleagues, previously demonstrated the BuChE inhibitory potential of the benzylpiperidine group which was synthesized from a series of cinnamic acid hybrids. These compounds were evaluated for their ability to inhibit both AChE and BuChE enzymes using the Ellman method with donepezil as the reference compound. These hybrids, which combine benzylpiperidine with various cinnamic acids, did not show any significant activity against AChE. However, some of these hybrids showed potent inhibitory effects against BuChE.

The core structure of cinnamic acid derivative (compound XIV—Figure 11) showed a moderate activity against BuChE, while the replacement of cinnamic acid with various other acids did not have a significant effect on the activity. In contrast, substitution with ferulic acid, methacrylic acid and 3-phenylpropionic acid resulted in increased inhibitory activity. Reduction of the double bond within the acid structure resulted in decreased activity for all compounds except compound XVI. Removal of the olefin group also resulted in lower activity. Compounds XV and XVI were also evaluated for their inhibitory effects on human BuChE. They were identified as potent inhibitors. In addition, these compounds showed moderate to strong inhibition of MAO-B. Compound XVI exhibited antioxidant properties, while compound XV was able to inhibit self-mediated Aβ1-42 aggregation and showed anti-inflammatory properties.

### 2.4. β-Carboline Analogs

*β*-Carboline analogues have shown activity against Alzheimer’s disease (AD) through various mechanisms, including cholinesterase and monoamine oxidase (MAO) inhibition, anti-inflammatory activity and anti-Aβ accumulation activity [121,122,123,124,125]. However, the physicochemical properties of the bivalent β-carboline group are particularly poor. To address this, Liao, Q. et al. [126] synthesized and evaluated bivalent *β*-carboline molecules with cinnamic acids to improve their properties while retaining the beneficial anti-AD effects of both groups.

The compounds that showed greater inhibition of Aβ accumulation are shown in Figure 12. The inhibition assay used a thioflavin T (ThT) fluorescence assay with resveratrol as the reference compound [127,128]. These compounds showed robust inhibitory activity against the enzyme BuChE and high selectivity for this enzyme over AChE. In addition, these molecules showed excellent neuroprotective effects with low neurotoxicity. Importantly, oral administration of these compounds was able to restore cognitive function in Alzheimer’s mouse models.

### 2.5. Tryptamine Derivatives

Tryptamine, a chemical group that has proven to be an effective scaffold for the synthesis of anticholinergic agents [129], was used by Ghafary, S. et al. [130] to create hybrid molecules with cinnamic acid. The anticholinergic activity of these compounds was evaluated by AChE and BuChE inhibition using Ellman’s method [97], with donepezil as the reference substance. The simplest unsubstituted compound exhibited moderate activity against both enzymes. The introduction of chlorine as a substituent on the benzoyl ring of cinnamate at different positions influenced the activity against AChE, with the ortho position showing the highest activity, whereas fluorine substitution decreased it. Notably, the compound with chlorine substitution at the ortho position showed the highest activity against BuChE (compound XX—Figure 13). Compounds with fluorine substitution showed increased activity against BuChE. Methyl substitution at each of the three positions slightly increased activity against AChE and more significantly against BuChE. Nitro substitution at all three positions significantly decreased activity against AChE. Significantly, increased anti-AChE activity seems to result from methoxy substitution, whereas double methoxy substitution at meta and ortho positions gives the most active AChE inhibitor (compound XXI—Figure 13). This compound is ineffective at inhibiting BuChE and exhibiting selectivity for AChE. A neuroprotective assay was then performed on compound XXI by exposing the PC12 cell line to H_2_O_2_-induced damage. The ability of the compound to inhibit beta-secretase 1 was also tested using the reference compound OM99-2, a hexapeptide derivative (CAS number 314266-76-7). Compound XXI showed no significant neuroprotective activity and inhibited beta-secretase 1 less than the reference compound. The compound was also evaluated for inhibition of self-induced Aβ aggregation using the thioflavin T (ThT) fluorescence method, with donepezil and curcumin as a reference. The compound was shown to inhibit Aβ aggregation to a level comparable to donepezil.

### 2.6. Rivastigmine Derivatives

Chen, Z. et al. [131] carried out a synthesis and evaluation of cinnamate derivatives of rivastigmine (Figure 1), producing eight derivatives of hydroxycinnamic acids. The aim was to exploit the properties associated with this group, which is known for its demonstrated antioxidant, neuroprotective and Aβ accumulation inhibitory properties. The synthesized compounds were tested for their activity against AChE and BuChE enzymes, using donepezil and rivastigmine as reference compounds. Most of the compounds exhibited more potent inhibition of BuChE compared to rivastigmine, while maintaining or enhancing activity against AChE. Furthermore, all compounds were subjected to a DPPH radical scavenging assay, which demonstrated antioxidant properties for almost all derivatives. In addition, these compounds were evaluated to determine the rate of inhibition of Aβ_1-42_ self-aggregation. Of the eight compounds, three showed significant activity in this assay (Figure 14).

## 3. Discussion

Alzheimer’s disease is a multifactorial disease with a complex nature and an elusive etiology, posing significant challenges in the development of novel therapeutic interventions. Recent research efforts have focused on the synthesis of hybrid molecules that target multiple facets of the disease. This article highlights the considerable promise of cinnamic acid as a basic structure that can be combined with various groups that show activity against Alzheimer’s disease. Table 1 briefly outlines the most potential compound hybrids synthesized and their relevant activities. Each compound contains a cinnamic acid moiety and an additional different group, demonstrating the significant influence of cinnamic acid on their efficacy.

Furthermore, these compounds exhibit multifunctional activities in line with the multi-target strategy. In addition to their inhibition of AChE and BuChE enzymes, the majority exhibit Aβ accumulation inhibition, antioxidant, and neuroprotective properties. In addition, some compounds show selectivity against BuChE, a key criterion in the development of new therapeutics.

Analysis of the compounds shows that, regardless of the group attached to the cinnamic acid moiety, a methoxy substitution at the benzene para position of cinnamic acid contributes significantly to increased potency. In general, electron-withdrawing substituents show increased activity, as exemplified by -CF3, with the exception of fluorine, which has shown reduced activity against AChE in specific studies.

The data suggest that the selectivity of the enzymes depends on the volume of the molecule, due to different spatial requirements in their active sites. However, this aspect requires further investigation.

Overall, there is an urgent need for extensive research into cinnamic acid hybrids for the prevention and treatment of Alzheimer’s disease. Given their promise, in-depth structural analysis would greatly advance our understanding and application of these compounds.

## Figures and Tables

**Figure 1 ijms-25-00582-f001:**
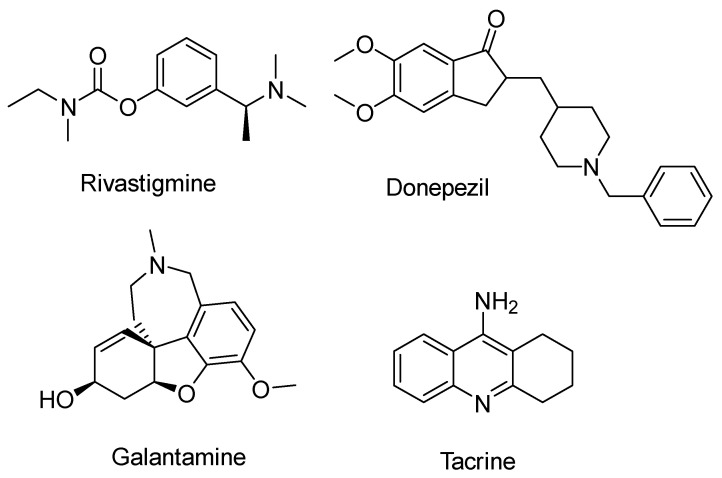
Chemical structures of the following compounds: rivastigmine, donepezil, galantamine and tacrine.

**Figure 2 ijms-25-00582-f002:**
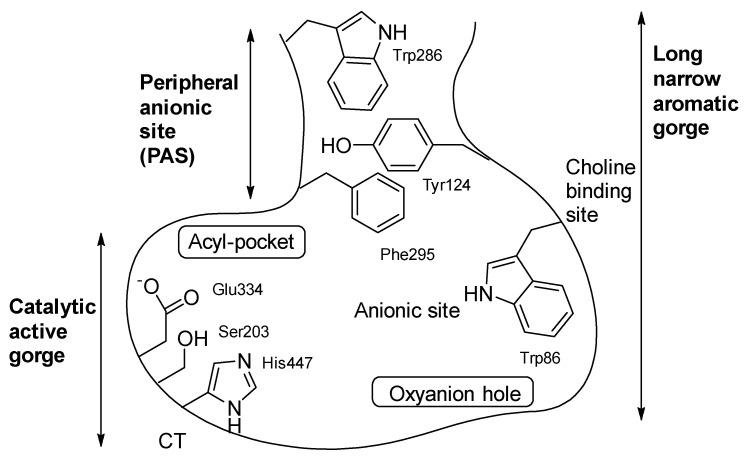
Two-dimensional representation of acetylcholinesterase showing the catalytic active site (CAS) located at the base of the enzyme and the peripheral anionic site (PAS) [56].

**Figure 3 ijms-25-00582-f003:**
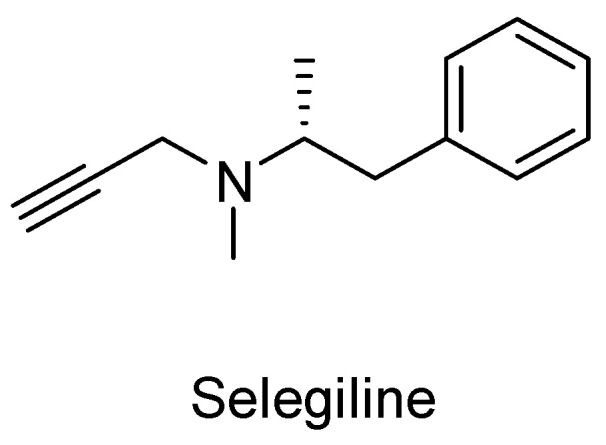
Chemical structure of selegiline.

**Figure 4 ijms-25-00582-f004:**
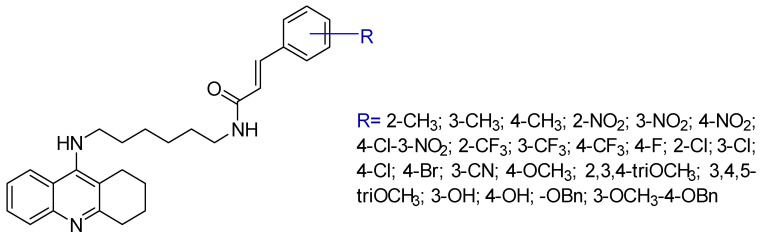
The compounds which were synthesized and examined by Chen, Y. et al. [96].

**Figure 5 ijms-25-00582-f005:**
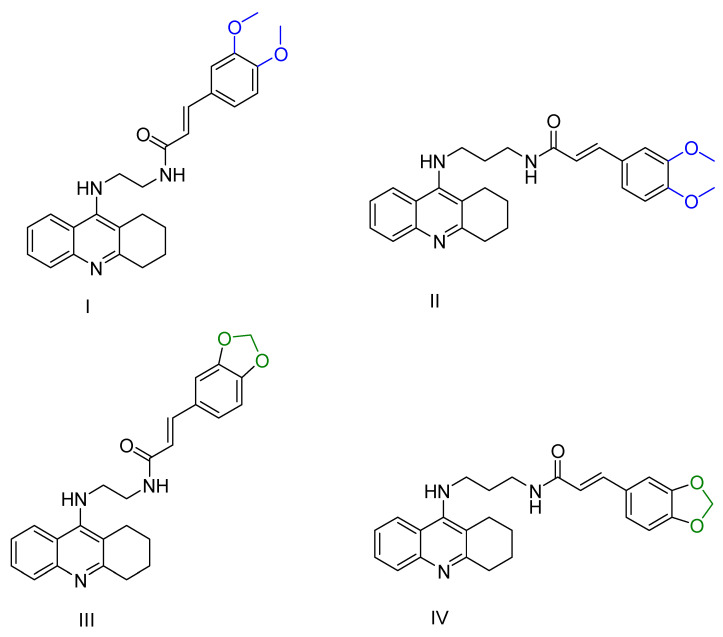
The compounds synthesized by Quintanova, C. et al. [100] being identified as the most promising.

**Figure 6 ijms-25-00582-f006:**
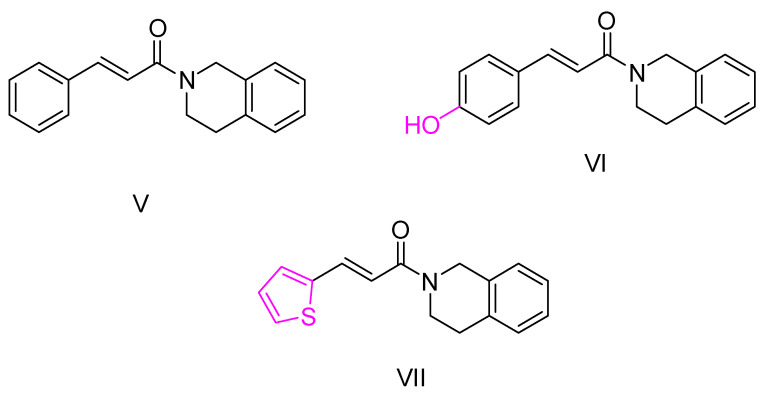
Compounds synthesized by Wang Κ. and colleagues [102]. Compound V is the unsubstituted skeleton of their molecules.

**Figure 7 ijms-25-00582-f007:**
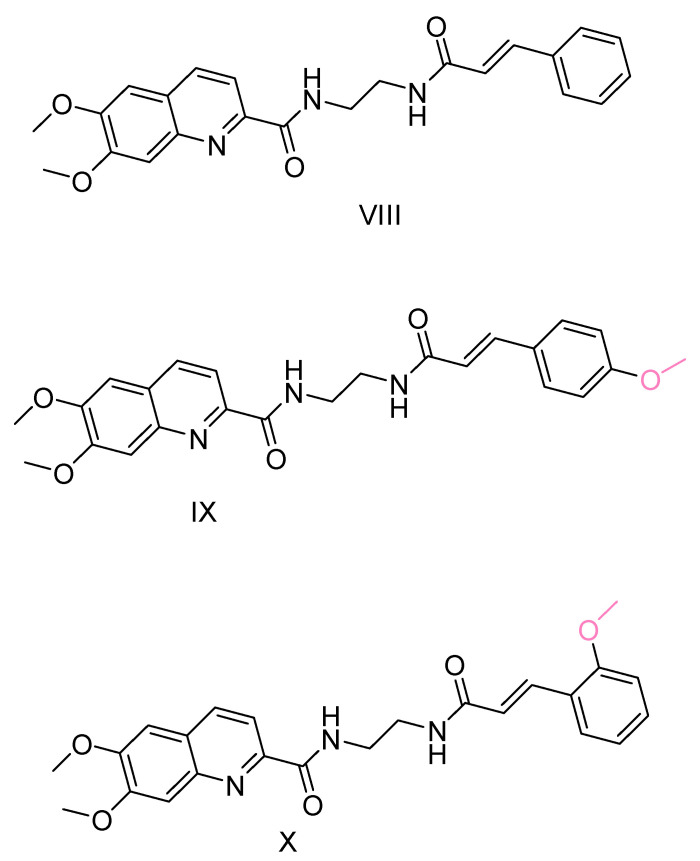
Compounds presenting most potential synthesized by Ge, Y-X. et al. [111].

**Figure 8 ijms-25-00582-f008:**
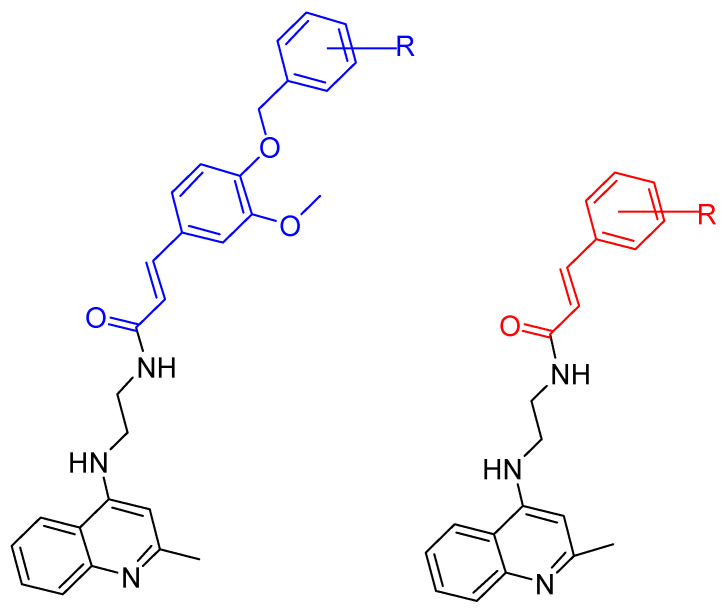
The basic molecular structure of the compounds synthesized by Mo, J. et al. [115]. The residual ferulic acid is shown in blue and the cinnamic acid in red.

**Figure 9 ijms-25-00582-f009:**
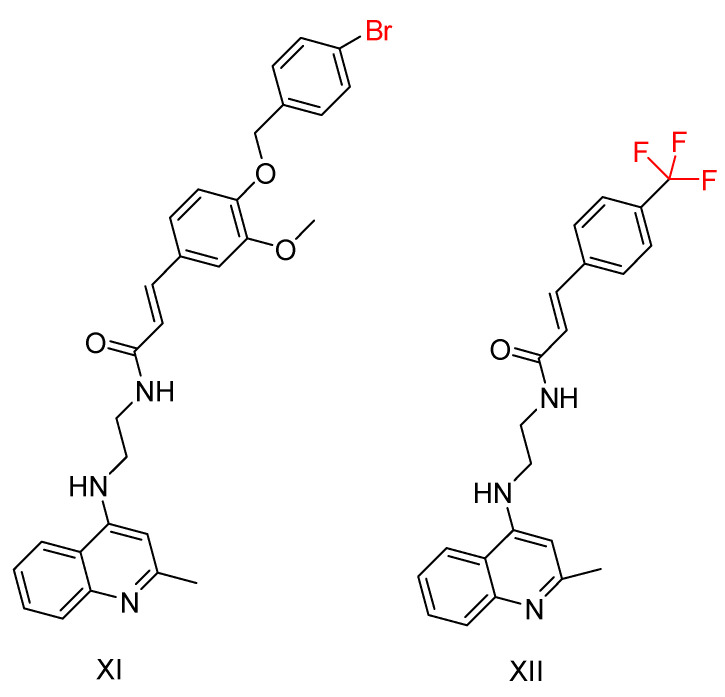
Compounds XI and XII are the most promising compounds synthesized by Mo, J. et al. [115].

**Figure 10 ijms-25-00582-f010:**
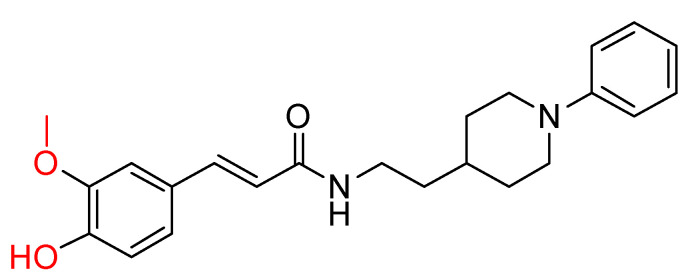
Compound XIII: the most active derivative synthesized by Estrada, Μ. et al. [120].

**Figure 11 ijms-25-00582-f011:**
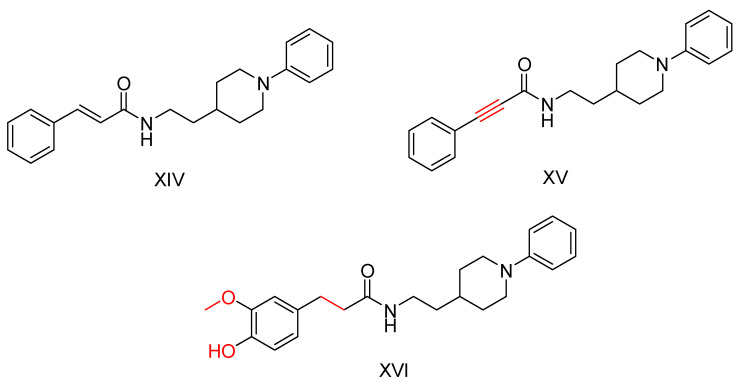
Compounds that have been synthesized by Wang, K. et al. [102]. Compound XIV is unsubstituted and acts as a reference compound while XV and XVI are the most active.

**Figure 12 ijms-25-00582-f012:**
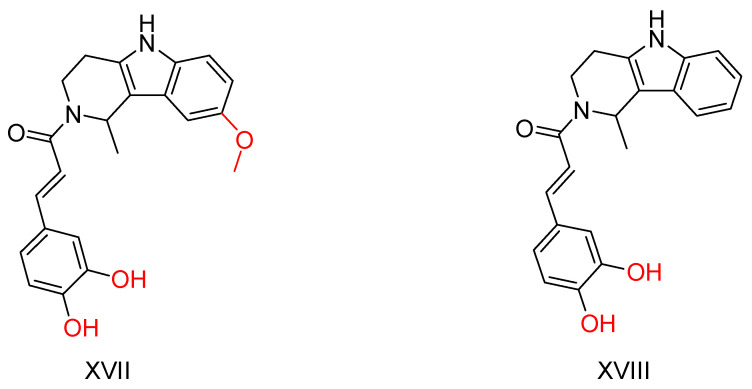
The most promising compounds synthesized by Liao, Q. et al. [126].

**Figure 13 ijms-25-00582-f013:**
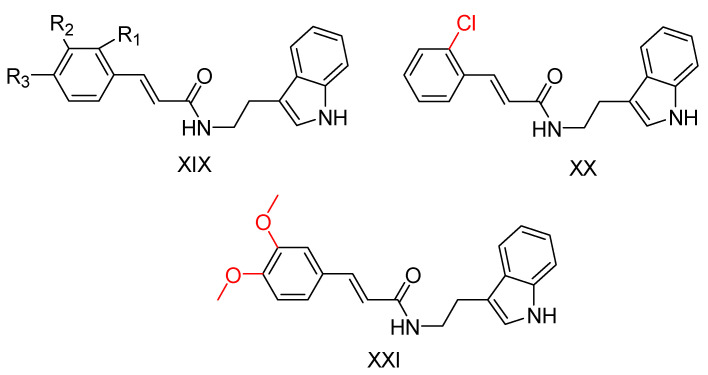
Compound XIX represents the general structure of compounds synthesized by Ghafary, S. et al. [130]. Among them, compound XX exhibited the highest activity as a BuChE inhibitor, whereas compound XXI demonstrated superior activity as an AChE inhibitor.

**Figure 14 ijms-25-00582-f014:**
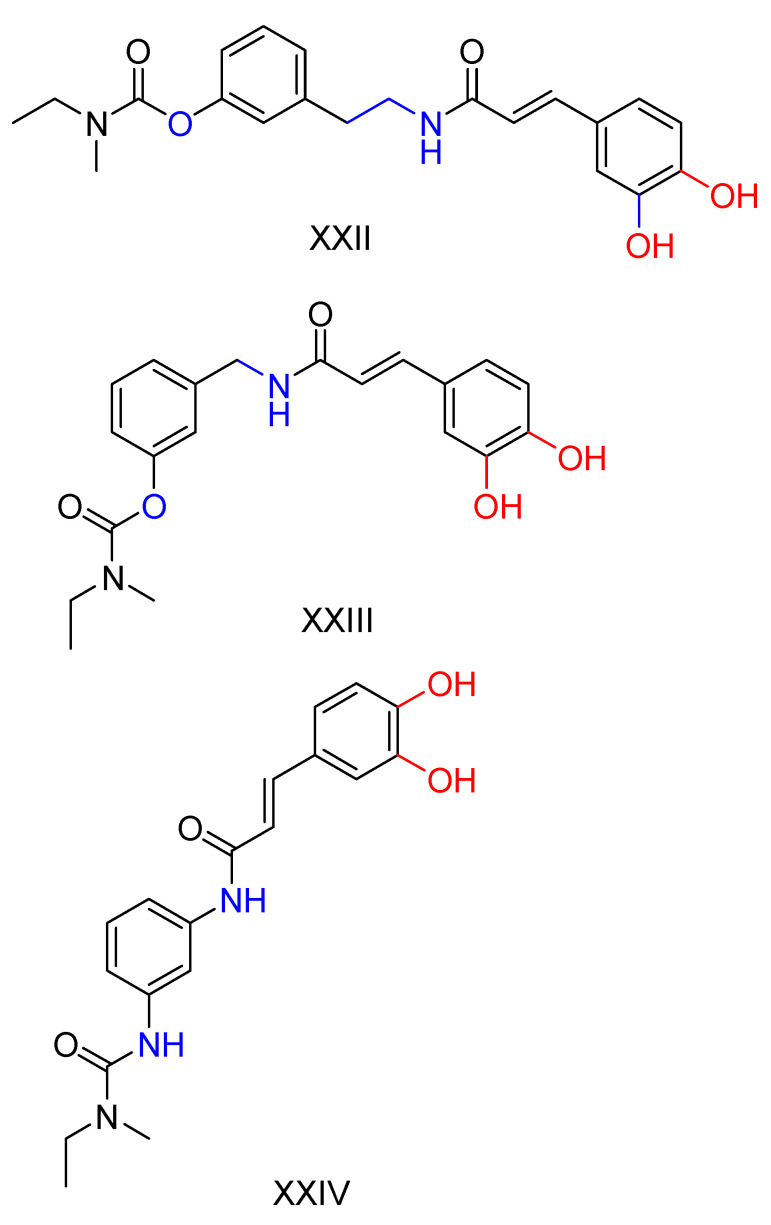
Three compounds that showed significant inhibition of Aβ accumulation [131].

**Table 1 ijms-25-00582-t001:** Summary of the most promising compounds and their activities.

Compound	Synthesized by	^a^ AChE-IC_50_ (μM) or %	^b^ huAChE- IC_50_(μM)	^c^ BuChE-IC_50_ (μM) or %	^d^ hu-BuChE-IC_50_ (μM)	^e^ % Inhibition of Self-Induced Aβ	Other
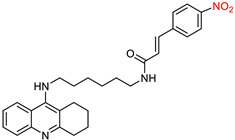	Chen, Y. et al. [96]	2.7 ± 0.4 nM	6.5 ± 0.6 nM	6.3 ± 0.3 nM	10.2 ± 1.2 nM	Under the concentration of 25 μM: 31.82	In vivo active (Morris water maze). Preliminary safe in hepatotoxicity studies.
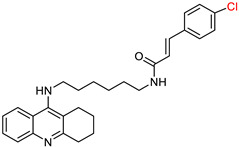	Chen, Y. et al. [96]	6.9 ± 1.2 nM	16.5 ± 1.7 nM	12.9 ± 1.7 nM	12.9 ± 1.7 nM	Under the concentration of 25 μM: 42.22	In vivo active (Morris water maze). Preliminary safe in hepatotoxicity studies.
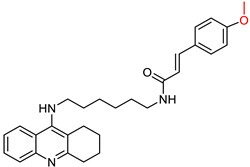	Chen, Y. et al. [96]	3.7 ± 1.5 nM	15.3 ± 1.8 nM	22.5 ± 5.9 nM	8.0 ± 1.1 nM	Under the concentration of 25 μM: 34.57	In vivo active (Morris water maze). Preliminary safe in hepatotoxicity studies.
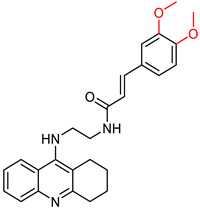	Quintanova, C. et al. [100]	0.09 ± 0.02	^f^ n.t.	n.t	n.t	Under the concentration of 80 μM: 19.6	Percentage of inhibition of antioxidant activity for 1 mM concentration (DPPH assay): 8.8%. 2.5μΜ of the compound showed protection against Aβ and H_2_O_2_ induced toxicity in neuroblastoma cells.
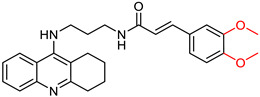	Quintanova, C. et al. [100]	0.09 ± 0.01	n.t.	n.t	n.t	Under the concentration of 80 μM: 56.5	-
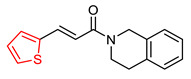	Wang, K. et al. [102]	19.4 ± 1.5	n.t.	2.1 ± 0.08	2.5 ± 0.09	Under the concentration of 25 μM: 65.2	Neuroprotective effects against Aβ-induced SH-SY5Y cell toxicity. Anti-inflammatory property (MTT assay). ^g^ MAO-B inhibition activity (IC_50_ = 1.3 μM). The compound improved dyskinesia recovery rate and memory impairment.
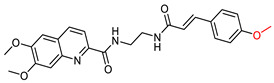	Ge, Y-X. et al. [111]	n.t.	n.t.	n.t.	n.t.	Under the concentration of 100 μM: 76.3 ± 2.6	Dose-dependent neuroprotective effect (evaluation) against Aβ_42_-induced neurotoxicity in SH-SY5Y cells.
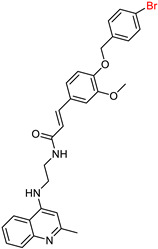	Mo, J. et al. [115]	3.54 ± 1.12	n.t.	1.51 ± 0.28	n.t.	No action	Neuroprotective effect against H_2_O_2_-induced cell death in PC12 neurons.
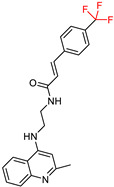	Mo, J. et al. [115]	3.28 ± 1.32	n.t	0.93 ± 0.40	n.t.	No action	Lower hepatotoxicity than tacrine.
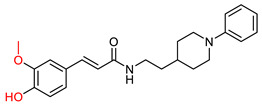	Estrada, M. et al. [120]	n.t.	0.39 ± 0.05	n.t.	0.076 ± 0.01	n.t.	Antioxidant properties (ORAC assay). Neuroprotective properties. (Protection of cell line SH-SY5Y against damage by free radicals in the mitochondria.)
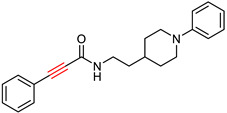	Wang, K. et al. [102]	16.5 ± 1.2	n.t.	3.3 ± 0.2	5.9 ± 0.4	Under the concentration of 25 μM: 70.3 ± 5.3	Anti-inflammatory properties. IC_50_ values against huMAO-B: 9.1 ± 0.2.
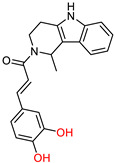	Liao, Q. et al. [126]	21.29 ± 3.63	n.t.	1.32 ± 0.85	n.t.	Under the concentration of 20 μM: 72.51 ± 0.84	Capability of restoring learning and memory function to AD model mice.
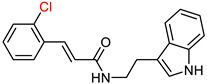	Ghafary, S. et al. [130]	21.11	n.t.	1.95	n.t.	n.t.	Selectivity over BuChE.
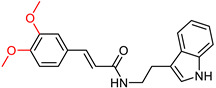	Ghafary, S. et al. [130]	11.51	n.t.	>100	n.t.	Under the concentration of 50 μM: 10.12 ± 1.23%	Mild neuroprotective activity.
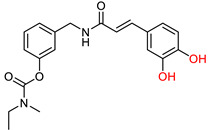	Chen, Z. et al. [131]	Under the concentration of 5 μΜ: 23.3%	n.t.	Under the concentration of 5 μΜ: 98.34%	n.t.	Under the concentration of 10 μM: 77.6%	Antioxidant properties (DPPH assay).

^a^ AChE from *Electric eel*, Ellman’s method; ^b^ AChE from human, Ellman’s method; ^c^ BuChE from equine serum, Ellman’s method; ^d^ BuChE from human, Ellman’s method; ^e^ inhibition of self-mediated Aβ_42_ aggregation (%) through the thioflavin-T fluorescence method; ^f^ not tested; and ^g^ recombinant human enzyme.

## Data Availability

There is no data to be reported in this study.

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
