# Peer review of "Multi-Target-Directed Cinnamic Acid Hybrids Targeting Alzheimer’s Disease"

_ijms, 2024, doi:10.3390/ijms25010582_

Round 1
Reviewer 1 Report
Comments and Suggestions for Authors
The article “Multi-target-directed cinnamic acid hybrids targeting Alzheimer’s disease” is a very well-written and organized review article that covers the topic adequately.
I understand that to achieve this order, the compounds have been re-numbered from previous versions. This led to some compounds being named as A, B, C, D, which does not correspond to the Roman numeral nomenclature that most of the compounds in the work have. Although the compounds in Figure 1 are referred to by letters, I understand that the compounds I am referring to do not refer to the compounds in Figure 1.
See lines 203, 210, 214 and 262.
Another small detail to improve is that in the summary the acronym Ab was left undefined (line 15).
Otherwise, I find the work very interesting and constitutes a very useful compilation for medicinal chemistry.
For all of the above, I recommend that the editor publish this work once these small details have been solved.
Author Response
We really thank the reviewer for its fruitful remarks. All the bellow mentioned corrections have been done as the reviewer suggested.The article “Multi-target-directed cinnamic acid hybrids targeting Alzheimer’s disease” is a very well-written and organized review article that covers the topic adequately.I understand that to achieve this order, the compounds have been re-numbered from previous versions. This led to some compounds being named as A, B, C, D, which does not correspond to the Roman numeral nomenclature that most of the compounds in the work have. Although the compounds in Figure 1 are referred to by letters, I understand that the compounds I am referring to do not refer to the compounds in Figure 1.
See lines 203, 210, 214 and 262.
Thank you for noticing the wrong numbering. We have foreseen and it has been corrected.
Another small detail to improve is that in the summary the acronym Ab was left undefined (line 15).
It has been corrected to Aβ.
Otherwise, I find the work very interesting and constitutes a very useful compilation for medicinal chemistry.
For all of the above, I recommend that the editor publish this work once these small details have been solved.
Reviewer 2 Report
Comments and Suggestions for Authors
This review intends to highlight the diverse actions of studied compounds as multi-target agents, analyze the outcomes of these studies, and summarize recent developments in the use of cinnamic acid derivatives synthesized by hybridization with proven bioactive moieties for the treatment of Alzheimer's disease.
The subject is relevant now because Alzheimer's disease, the most frequent mental illness that strikes people in their later years, is predicted to rise in frequency over the next several years.
The writing is methodical and really well-organized and the literature is extensive, comprehensive and current.
In my opinion, it is not necessary to mark the residues of ferulic and cinnamic acid in Figure 18, and associated functional groups are not colored in Figures 13 and 14.
I think that this paper is well written, systematically reviewed, and comprehensively covers the stated research topic, and I recommend it to be published with minor corrections.
Author Response
We really appreciate the reviewers’ fruitful comments and the time spent for the reviewing process in order to improve our manuscript.
This review intends to highlight the diverse actions of studied compounds as multi-target agents, analyze the outcomes of these studies, and summarize recent developments in the use of cinnamic acid derivatives synthesized by hybridization with proven bioactive moieties for the treatment of Alzheimer's disease.
The subject is relevant now because Alzheimer's disease, the most frequent mental illness that strikes people in their later years, is predicted to rise in frequency over the next several years.
The writing is methodical and really well-organized and the literature is extensive, comprehensive and current.
In my opinion, it is not necessary to mark the residues of ferulic and cinnamic acid in Figure 18, and associated functional groups are not colored in Figures 13 and 14.
I think that this paper is well written, systematically reviewed, and comprehensively covers the stated research topic, and I recommend it to be published with minor corrections.
It must be a mistyping somewhere because there is no Figure 18 in the manuscript. By coloring certain groups, we aim to highlight the differences between the functional groups of different series. We have added color in Figures 13 and 14 so as to be the same as the other ones. The other option is to remove all the coloring from all the compounds and figures if the reviewer’s thinks it is more enlightening.
Reviewer 3 Report
Comments and Suggestions for Authors
Review Report on Dec6,2023
Multi-target-directed cinnamic acid hybrids targeting Alzheimer’s disease
Key points to be answered:
· Abstract should be more elaborated with the research results.
· Sec. 2.1, [To achieve this, a series of tacrine-based molecules were synthesized, each attached to cinnamic acid via a diamine linker chain, varying in the number of intermediate carbons. Compounds with a higher number of intermediate carbons, especially those with 6 and 8 carbon atoms, showed greater efficacy against AChE. Among these two compounds, the one with a 6-carbon linker was the most favorable due to its comparable activity and lower molecular weight.] However, authors have not produced the binding studies?
2.2 Isoquinoline & Quinoline derivatives
· [Building on their previous research, Wang K. and colleagues [81], who had previously demonstrated the inhibitory potential of the 1, 2, 3, 4-tetrahydroisoquinoline group against the BuChE enzyme [82], developed hybrid compounds by combining this group with various cinnamic acids. These compounds were screened as inhibitors of AChE, and BuChE enzymes using the Ellman method with donepezil as the reference compound.] More references should be cited.
· Authors should cite more relevant references in the introduction part since Alzemer’s disease having lot of history since 2 decades.
· Authors should provide the structures of docking images with the potential inhibitors for easy understanding of journal readers.
· I found regular missing of spaces between words, comas, and full stops including sub-headings in the entire draft.
· The overall review article is well executed and written, this review is suitable for the publication after minor revision.
·
I recommend this article for minor revision after fixing of above points.
Regards
Comments on the Quality of English LanguageReview Report on Dec6,2023
Multi-target-directed cinnamic acid hybrids targeting Alzheimer’s disease
Key points to be answered:
· Abstract should be more elaborated with the research results.
· Sec. 2.1, [To achieve this, a series of tacrine-based molecules were synthesized, each attached to cinnamic acid via a diamine linker chain, varying in the number of intermediate carbons. Compounds with a higher number of intermediate carbons, especially those with 6 and 8 carbon atoms, showed greater efficacy against AChE. Among these two compounds, the one with a 6-carbon linker was the most favorable due to its comparable activity and lower molecular weight.] However, authors have not produced the binding studies?
2.2 Isoquinoline & Quinoline derivatives
· [Building on their previous research, Wang K. and colleagues [81], who had previously demonstrated the inhibitory potential of the 1, 2, 3, 4-tetrahydroisoquinoline group against the BuChE enzyme [82], developed hybrid compounds by combining this group with various cinnamic acids. These compounds were screened as inhibitors of AChE, and BuChE enzymes using the Ellman method with donepezil as the reference compound.] More references should be cited.
· Authors should cite more relevant references in the introduction part since Alzemer’s disease having lot of history since 2 decades.
· Authors should provide the structures of docking images with the potential inhibitors for easy understanding of journal readers.
· I found regular missing of spaces between words, comas, and full stops including sub-headings in the entire draft.
· The overall review article is well executed and written, this review is suitable for the publication after minor revision.
·
I recommend this article for minor revision after fixing of above points.
Regards
Author Response
Thank you for all the fruitful suggestions and comments aiming to improve our manuscript. We have tried to explain and address all the reviewers’ comments.
Multi-target-directed cinnamic acid hybrids targeting Alzheimer’s disease.
Key points to be answered:
- Abstract should be more elaborated with the research results.
The results have been elaborated in the abstract as the reviewer suggested.
- Sec. 2.1, [To achieve this, a series of tacrine-based molecules were synthesized, each attached to cinnamic acid via a diamine linker chain, varying in the number of intermediate carbons. Compounds with a higher number of intermediate carbons, especially those with 6 and 8 carbon atoms, showed greater efficacy against AChE. Among these two compounds, the one with a 6-carbon linker was the most favorable due to its comparable activity and lower molecular weight.] However, authors have not produced the binding studies?
This conclusion about the importance of the 6-8 carbon atoms is not based on docking studies but in biological results. The authors of the original paper have not produced the docking studies. As mentioned by Chen, Y. et al., firstly they have tried to explore the importance of the linker between tacrine and cinnamic acid. These biological results showed that the presence of a 2-4 carbon linker to the diamine does not favor the inhibitory activity and a 6-8 carbon linker has proved to be the optimal for the activity. We have tried to clarify it in this part in the manuscript.
2.2 Isoquinoline & Quinoline derivatives
- [Building on their previous research, Wang K. and colleagues [81], who had previously demonstrated the inhibitory potential of the 1, 2, 3, 4-tetrahydroisoquinoline group against the BuChE enzyme [82], developed hybrid compounds by combining this group with various cinnamic acids. These compounds were screened as inhibitors of AChE, and BuChE enzymes using the Ellman method with donepezil as the reference compound.] More references should be cited.
Additional references have been added as the reviewer suggested.
- Authors should cite more relevant references in the introduction part since Alzheimer’s disease having lot of history since 2 decades.
More relevant references have been added.
- Authors should provide the structures of docking images with the potential inhibitors for easy understanding of journal readers.
It is worth mentioning that even all the original papers do not include structures of docking images with the potential inhibitors. Besides that, this review article aims to collect all the recent findings on cinnamic acid hybrids as promising therapeutic candidates for the treatment of AD and not to explore their potential binding mode.
- I found regular missing of spaces between words, comas, and full stops including sub-headings in the entire draft.
We have checked the entire manuscript and all the punctuation errors have been corrected.
- The overall review article is well executed and written, this review is suitable for the publication after minor revision.
I recommend this article for minor revision after fixing of above points.